# Conformation of the Intermediates in the Reaction Catalyzed by Protoporphyrinogen Oxidase: An In Silico Analysis

**DOI:** 10.3390/ijms21249495

**Published:** 2020-12-14

**Authors:** Abigail L. Barker, Hamlin Barnes, Franck E. Dayan

**Affiliations:** Agricultural Biology Department, Colorado State University, Fort Collins, CO 80523, USA; abigail.barker@colostate.edu (A.L.B.); ruth.enocene@yahoo.com (H.B.)

**Keywords:** reaction mechanism, docking of tetrapyrrole macrocycles, Lennard-Jones forces

## Abstract

Protoporphyrinogen oxidase (PPO) is a critical enzyme across life as the last common step in the synthesis of many metalloporphyrins. The reaction mechanism of PPO was assessed in silico and the unstructured loop near the binding pocket was investigated. The substrate, intermediates, and product were docked in the catalytic domain of PPO using a modified Autodock method, introducing flexibility in the macrocycles. Sixteen PPO protein sequences across phyla were aligned and analyzed with Phyre2 and ProteinPredict to study the unstructured loop from residue 204–210 in the *H. sapiens* structure. Docking of the substrate, intermediates, and product all resulted in negative binding energies, though the substrate had a lower energy than the others by 40%. The α-H of C10 was found to be 1.4 angstroms closer to FAD than the β-H, explaining previous reports of the reaction occurring on the *meso* face of the substrate. A lack of homology in sequence or length in the unstructured loop indicates a lack of function for the protein reaction. This docking study supports a reaction mechanism proposed previously whereby all hydride abstractions occur on the C10 of the tetrapyrrole followed by tautomeric rearrangement to prepare the intermediate for the next reaction.

## 1. Introduction

The porphyrin pathway plays a central role in the synthesis of many pigments fundamental to sustaining life across the biological realm [1,2,3]. Macrocyclic tetrapyrroles derived for this pathway have a varying degree of conjugation and can be chelated with metal dications such as the alkaline earth metal Mg^2+^, transition metals such as Fe^2+^, Co^2+^, and Ni^2+^, and group 12 element Zn^2+^. These metals are coordinated with the pyrrole nitrogens at the center of these rings. Examples of biologically important metalloporphyrins include chlorophylls (Mg^2+^), hemes (Fe^2+^), vitamin B12 (Co^2+^), and coenzyme F430 (Ni^2+^).

Biosynthesis of porphyrins consists of 7 core enzymes that culminates in the action catalyzed by protoporphyrinogen oxidase (PPO) converting the colorless precursor protoporphyrinogen IX (protogen) into the fully conjugated, bright red pigment protoporphyrin IX (proto) [3]. PPO itself is of particular interest because mutations of the protein lead to variegate porphyria diseases in humans and in agriculture it is the protein target of the PPO-inhibiting herbicides which are seeing a resurgence of use in recent years [4,5].

There have been many inquiries into the mechanism of the three sequential oxidation reactions which convert protogen into the fully conjugated proto. An early study of the biosynthesis of heme using isotopically labelled tetrapyrrole determined that the reaction involved hydride abstractions from three of bridging methylene groups [6]. This mechanism posited two significant mechanistic properties: the hydride ion was abstracted from the *meso* face (α-H) and tautomerism around the neighboring tetrapyrrole ring resulted in the formation of a double bond (Figure 1). 

This mechanism required that protogen and the intermediates rotate within the catalytic domain so that the *meso* face of each methylene functionality was exposed to FAD for the next hydride abstraction to occur. However, elucidation of PPO’s structure by X-ray crystallography suggested that protogen is held in a relatively fixed position within the catalytic domain by having the propionate group from ring D (Figure 1) interacting with the guanidino functional group of Arg97 and maintaining the C10 carbon of protogen in close proximity to N5 of the cofactor FAD [7]. Still, no published crystal structure of PPO has been reported with either the substrate or the product in the active site. A previous effort to bind protogen to PPO in silico was performed with the structure of protogen created by an extensive analysis process and a structure of PPO which was allowed to relax in a molecular dynamics simulation [9]. This method does not take into account the structure of the precursors in the pathway which are unlikely to change due to the repulsion of the hydrogen molecules in the center of the tetrapyrrole ring, and new methods with the Autodock program have simplified the analysis of flexible substrates such as protogen. 

The most current understanding of PPO’s reaction mechanism involves 3 subsequent hydride abstractions on C10 of protogen, the methylene bridge between pyrrole rings B and C, followed by tautomerization to regenerate the unsaturated state of C10 allowing for the next reaction [7]. Mechanistically, the first reaction is initiated by a 1:1 stoichiometric FAD as a coenzyme. The methylene functionality is prochiral because the α-H is much closer to N5 of FAD than the β-H (Figure 1). Once the *meso* hydrogen is removed in the form of a hydride ion, it creates an unstable carbocation transition state at C10, which instantly rearranges in ring B as shown in intermediate 1a in Figure 1. The second reaction is an enamine-imine tautomerization from 1b to 1c resulting in the reformation of the methylene bridge at C10 by transfer of the proton from ring B pyrrole nitrogen. FADH^−^ is oxidized to FAD by reaction with O_2_ and the proton released from pyrrole N in the third reaction and generates H_2_O_2_. The second step repeats the same sequence of reactions. Oxidation of C10 by FAD, 1c and instant rearrangement of ring C, 2a followed by the enamine-imine tautomerization 2b to 2c and the subsequent reoxidation of FADH^−^ to FAD. The third step involves the oxidation of C10 by FAD and four two electron rearrangements from ring D, 2d to yield proto. 

Multiple crystal structures of PPO from differing organisms show similar folding despite a large variation in conservation of sequence: from around 90% identity between mammals to as low as 23% identity between mammalian and plant sequences (Appendix A). There are two separate isoforms of PPO in plant species which only share about 27% sequence identity, generally considered the chloroplastic and mitochondrial isoforms. Along with the mostly conserved structure, some conserved residues across phyla include the Arg97 region and the Phe353-Leu356 region [4]. One conserved feature is an unstructured loop that is not captured in any of the published crystal structures that is adjacent to the catalytic domain of the protein, located from residue 204 to 210 in the human PPO. The flexible nature of the loop and proximity to the reaction pocket could signify importance if these sequences are conserved across phyla. 

We aim to better understand the reaction catalyzed by PPO by docking the substrate and product of the reaction along with relevant reaction intermediates, describe the reaction mechanism in a way that unifies previous data while taking into account the spatial limitation of the catalytic domain of PPO as well as information from substrate orientations upstream in the pathway, and to determine if sequence similarity suggests that the unstructured loop has a function in the reaction or substrate binding. 

## 2. Results and Discussion

### 2.1. Determining the Initial Conformation of Protogen

Protogen is a very flexible tetrapyrrole because each pyrrole ring is connected to the other via unsaturated methylene bridges. There is no published crystal structure of protogen cocrystallized with any protein, and consequently it is difficult to predict the conformation of protogen in PPO. Free protogen creates a dramatically bent shape naturally because of the repelling forces between the center hydrogen atoms. This natural shape does not fit in the catalytic domain of PPO. Furthermore, there is evidence of protein channeling or chaperoning of the intermediates in this pathway [7,10,11]. Therefore, we obtained a realistic starting conformation from the structure of coproporphyrinogen III (copro), which contains the same repelling hydrogens and flexible methylene bridges, cocrystallized with uroporphyrinogen decarboxylase (1R3Y) [8]. Since the only difference between copro and protogen is the presence of propionic acids instead of vinyl groups on rings A and B (Figure 2), protogen was constructed from the coordinate of the optimized structure of copro. This slightly concave conformation is a particularly good starting point for docking protogen and the reaction intermediates because it fits relatively well in the catalytic domain of PPO. 

The structures of all the subsequent reaction intermediates were derived from this original confirmation by editing the mol2 files followed by geometric optimization with Spartan18 according to the putative reaction scheme proposed by Koch et al. [7] (Figure 1). 

### 2.2. Docking of Protogen, Tautomeric Reaction Intermediates and Proto to PPO

The reaction catalyzed by PPO involves 3 oxidative steps on the same carbon (C10) forming the methylene bridge between the B and C pyrrole rings followed by tautomeric rearrangements (Figure 1). In order to dock protogen and all the relevant reaction intermediates in PPO, the docking protocol for Autodock was modified to allow flexibility of the tetrapyrrole macrocycles according to the method developed by Forli [12,13]. The method consists of ‘breaking’ one of the bonds in the tetrapyrrole rings, replacing carbons at dummy atoms and applying a Lennard-Jones force between these dummy atoms to keep them proximal to each other during the docking procedure. Accordingly, we avoided bonds in partially rigid regions to reduce the complexity and calculation time without compromising accuracy. Consequently, the bond between the methylene carbon between rings C and D was selected (Figure 3) since it is the most flexible bond throughout all the steps shown in Figure 1, due to the fact that it is the last single bond to be oxidized. This method creates a flexible ring system with an increased number of rotatable bonds to be considered during calculations.

Proto was obtained from x-ray analysis of protoporphyrin IX dimethyl ester by removing the methyl esters. The fully conjugated product of the reaction catalyzed by PPO is known to be a non-flexible planar structure and was docked in PPO without breaking the cyclic tetrapyrrole. The location of the catalytic domain was defined using a gridbox that encompassed the FAD forming the ‘roof’ of the cavity and, the top of α-8 helix of PPO upon which the tetrapyrrole rings are centered at the ‘bottom’ of the cavity, and Arg97 involved in stabilizing the rings on the right side of the cavity (Appendix A).

After the docking, RMSD of the clusters with best poses for each tetrapyrrole intermediate provide insight to the changes in the curvature of the tetrapyrroles during the oxidation between protogen and proto (Figure 4). The RMSD for protogen is small, relative to the other reaction intermediates 1a, 1c, 2a and 2d. The low RMSD of protogen is most likely a reflection of the flexibility afforded by the fully unsaturated methylene bridges linking the pyrrole rings. In fact, the movement of the propionate side chains probably accounts for a significant portion of the RMSD. On the other hand, the increased RMSD of the reaction intermediates is due to the increased rigidity and planarity of the macromolecules as additional double bonds are introduced in the methylene bridges. The RMSD of the proto is also low. It should be noted that this refers to the coordinates of the crystal structure published by Caughey and Ibers [14]. Since proto retained its planar conformation relative to the starting conformer used for docking, little change in the shape of the tetrapyrrole ring is expected. As with protogen, a significant portion of the RMSD is probably due the movement of the propionate side chains.

When docked in the catalytic domain the binding energy of protogen averaged −10.59 ± 1.08 kcal/mol (Figure 5). The binding energies of all reaction intermediates and proto were on average 40% greater than that of protogen, ranging from −6.00 to −7.27 kcal/mol. This is accounted for by the fact that protogen is the most flexible of the tetrapyrroles, having 4 saturated methylene bridges connecting each pyrrole rings. All subsequent tetrapyrroles are less flexible because of the introduction of an increasing number of unsaturated methylene bridges, leading to the fully conjugated and least flexible product of the reaction (proto). These binding energies agree with the paradigm of PPO-protogen > PPO-proto established in Hao et al. [9].

The docking protocol allowed for the propionate groups of either ring C or D to interact with the guanidino functional group of Arg97 (Figure 6), all the docked molecules from cluster 1 assumed the expected pose with the propionate group from ring D interacting with and maintaining the C10 carbon of protogen in close proximity to N5 of the cofactor FAD (Figure 6) [7]. On average the propionate group was between 4.3 and 4.6 Å from Arg 97 and the C10 of protogen (and the reaction intermediates) was between 3.3 and 4.1 Å from N5 of FAD (Table 1). 

A key feature of the reaction postulated by Koch et al. [7] is that the oxidation of protogen to proto involves three sequential hydride abstraction from C10 of the substrate and subsequent reaction intermediate catalyzed at the N5 of FAD. Our docking study supports this by consistently positioning the α-H from the C10 of the tetrapyrroles facing the re-face of N5 of FAD. Furthermore, the average distance between N5 and α-H range from 3.6 to 3.7 Å (Table 2), which is within the known range for redox reactions catalyzed by FAD enzymes [15,16,17,18,19,20,21]. On the other hand, the β-H are on average 1.4 Å further from the N5 of FAD (range from 5.0–5.1 Å). The second step requires tautomerization with an adjacent pyrrole ring and subsequent loss of a proton from the pyrrole nitrogen, to regenerate the sp3 carbon at C10 prior to the next hydride abstraction. Experiments using isotopically labeled substrates suggested that the α-Hs were preferentially removed during the oxidation of protogen into proto (Figure 1) [6]. Docking of protogen and the reaction intermediates revealed that the poses with the lowest binding energy had the tetrapyrrole rings in a slightly concave conformation for optimal fit within the catalytic domain of PPO (Figure 6). In this conformation, the α-Hs are closer than the β-Hs to the adjacent pyrrole bond involved in the tautomerization (Table 2), which accounts for the preferential loss of the *meso* hydrogen reported by Jones et al. [6]. 

Hydride abstractions are involved in many enzymatic reactions [21,22,23]. However, the mechanisms involved in hydride transfers had never been demonstrated empirically in biological systems until the recent groundbreaking work on protochlorophyllide oxidoreductase [24]. While that study involves a tetrapyrrole molecule downstream from PPO, the enzyme being considered herein, the reaction is not identical. In the case of protochlorophyllide oxidoreductase, the cofactor is NADPH (instead of FAD), and the direction of the hydride transfer is from the cofactor to ring D of the substrate, rather than from the methylene bridge between ring B and C of the substrate to the cofactor. Nevertheless, this new understanding of hydride transfer mechanisms provides some relevant insight in the reaction catalyzed by PPO. Indeed, the participation of FAD in a wide range of redox reactions is well documented [23,25]. In particular, the N5 of the oxidized tricyclic isoalloxazine ring system is often a site for hydride addition, converting FAD to FADH^−^ [26]. 

### 2.3. Characterization of the Disordered Loop Crossing the Opening of PPO Binding Domain

Following the docking, an alignment was created of the top 80 related sequences to the human PPO sequence from the crystal structure used for this research to investigate the unstructured, flexible loop region near the catalytic domain, from residues 204–210. The flexibility and proximity could indicate that the loop has a function in the binding of the substrate or the release of the product. Residues that are conserved in homologous proteins between species and phyla are often associated with important function [27]. Alignment of the top 80 sequences showed a lack of homology from residue 205–209 which covers the entire sequence that is not captured in crystal structures. Furthermore, in the alignment of sequences across phyla the loop is not only not conserved at the residue level but varies in length as well (Figure 7). In protein structure predictions, across all species, the region is considered a solvent exposed region and a disordered region showing no defined secondary structure. Based on these observations, it is unlikely that the region is directly involved with the binding of the substrate or the reaction carried out by PPO. Conservation of the flexible region could indicate the necessity of movement near the catalytic domain to accommodate substrate and product movement. 

## 3. Conclusions

A novel docking model for protogen in the catalytic domain of PPO was created using a protogen structure derived from the crystal structure of a close precursor in the pathway, copro, bound to the catalytic domain of uroporphyrinogen decarboxylase. The intermediates in the pathway were created on the same backbone structure for docking, and the product proto was created based on previous crystallographic data. Binding energy for protogen was the most favorable with an average of −10.59 ± 1.08 kcal/mol, and the intermediates and product showed binding energies ~40% higher. The binding energies are consistent with the flexibility of the molecules, protogen being the most flexible and each intermediate thereafter reducing flexibility with the addition of double bonds. The docked molecules with the most energetically favorable positions all maintained an association between the propionate group of ring D and Arg98, which held the α-H of C10 within the appropriate range for redox reactions to occur with the N5 of FAD (3.6–3.8 Å). The findings here unify early experiments noting the hydride abstractions with the current understanding of the catalytic domain of PPO and known precursor conformations in the pathway. Finally, the unstructured loop near the opening to the catalytic domain was investigated for possible interaction with the substrate. A complete lack of homology, even within a phyla, indicates that no particular residue interacts with the substrate, but the conservation of the flexible region may indicate the necessity of accommodating substrate and product movement near the catalytic domain. 

## 4. Materials and Methods 

### 4.1. Determining the Initial Conformation of Protogen and Construction of All the Reaction Intermediates

Conformation of protogen was from derived from the structure of coproporphyrinogen III (copro) cocrystallized with uroporphyrinogen decarboxylase (1R3Y) [8]. This structure was extracted from the pdb file and saved as a mol2 file. As is common for ligand structures determined by protein crystallography, the atom types and bond orders of copro had to be reconstructed in silico using a molecular modelling and computational chemistry application (Spartan18, Wavefunction, Inc. Irvine, CA 92612). The bond angles and length were corrected by submitting the individual pyrrole rings to geometric minimization using density function theory calculations (wB97X-D 6-31*) (Appendix A). The propionic acid groups found on rings A and B of the tetrapyrrole were converted to vinyl groups to convert copro into protogen. Once protogen was constructed, all the reaction intermediates (1a, 1c, 2a, 2d) were built from this original confirmation by editing the mol2 files and submitted to geometric optimization with Spartan18. The structure of proto was obtained from the X-ray analysis of protoporphyrin IX dimethyl ester [14] deposited in The Cambridge Crystallographic Data Centre (CCDC) [28]. Spartan18 was used to remove the methyl esters and convert the propionic acids to propionate.

### 4.2. Docking of Protogen, Tautomeric Reaction Intermediates and Proto to PPO 

The crystal structure of human PPO was obtained from 3nks [29]. The atom types of FAD were corrected using Spartan 18 and the ligand was converted to its oxidized form. The electrostatic charges were calculated using density function theory calculations (wB97X-D 6-31*). 

Protogen and all of the reaction intermediates were docked into the catalytic domain of PPO using a modified Autodock method using Lennard-Jones potential and dummy atoms to overcome the software’s limitation (AutoDock version 4.2, Scripps Institute, San Diego CA, USA) in handling flexible ring systems (Figure 3) [12,13,30,31]. This was done by editing the ligand.pdbqt files. Additionally, the guanidino group of arginine 97 was designated as important in the interaction between one of the propionate groups and a grid box was used to delimitate the region of the catalytic domain according to the software [13]. The gridbox dimensions were set to 90 × 75 × 75 points with a spacing to 0.2. The box was centered on the following coordinates: x = −26.88, y = 4.47 and z = 45.17. PPO was set as a rigid structure (Appendix A). The cyclic tetrapyrroles modified to allow for flexible docking designed in the previous section were used for docking within the space defined in the gridbox. The algorithm was set to generate 50 docking poses and the top clustered was selected as optimal conformation for the docking of each ligand. 

### 4.3. Modeling of Disordered Loop Crossing the Opening of PPO Binding Domain

The human PPO sequence was blasted in the NCBI database and the top 80 related sequences were selected. A second data set of 16 high quality PPO sequences spanning multiple phyla were selected to determine base homology (Appendix A). The two datasets were aligned using Clustal Omega [32]. The loop sequences from the more diverse dataset were compared from the alignment and then run through two predictive structure programs: Phyre2 [33] and PredictProtein [34]. 

## Figures and Tables

**Figure 1 ijms-21-09495-f001:**
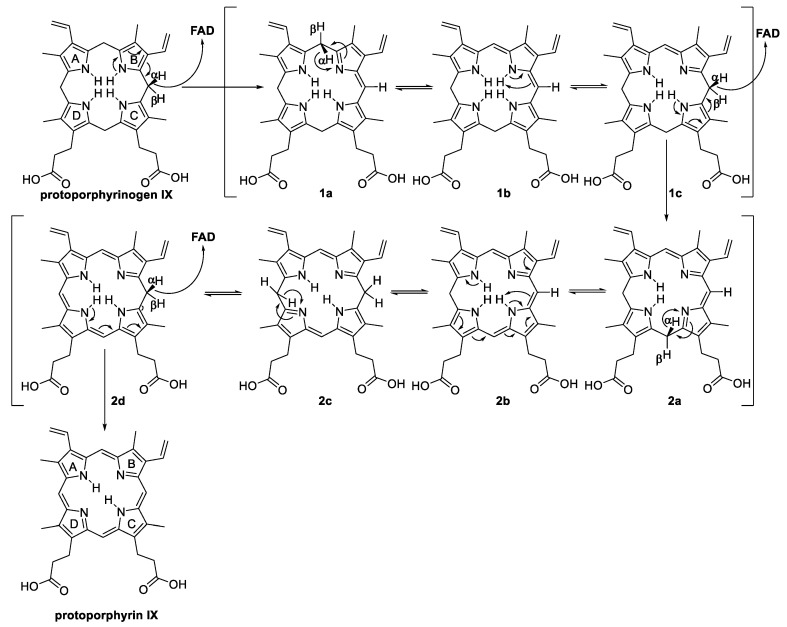
Putative reaction intermediates involved in the catalytic reaction of PPO converting protogen and proto. This figure is adapted from reaction scheme proposed by Koch et al. [7] but orienting the tetrapyrrole ring to match that of copro in its binding site in uroporphyrinogen decarboxylase (1R3Y) [8].

**Figure 2 ijms-21-09495-f002:**
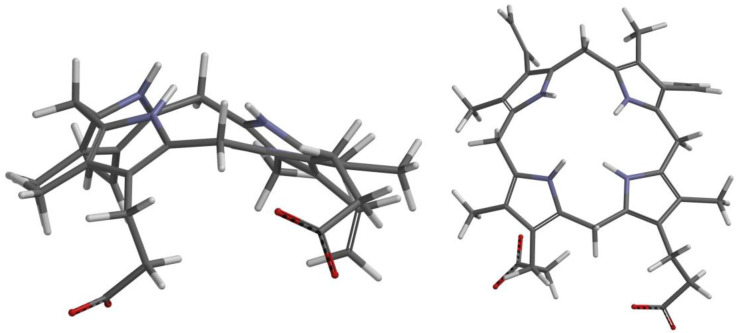
Starting conformation of geometrically optimized protogen based on the coordinates of copro in uroporphyrinogen decarboxylase (1R3Y) [8].

**Figure 3 ijms-21-09495-f003:**
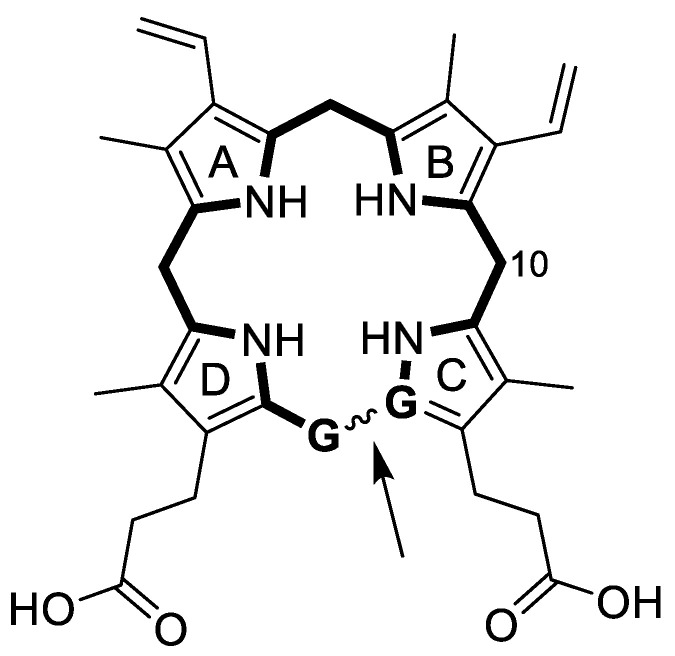
Localization of the broken bond (arrow) and respective dummy atoms (labeled **G**) selected to allow docking of the flexible ring system of the tetrapyrrole intermediates in the conversion of protogen to the last tautomer before proto. The carbon (C10) where the hydride transfers occur is labelled.

**Figure 4 ijms-21-09495-f004:**
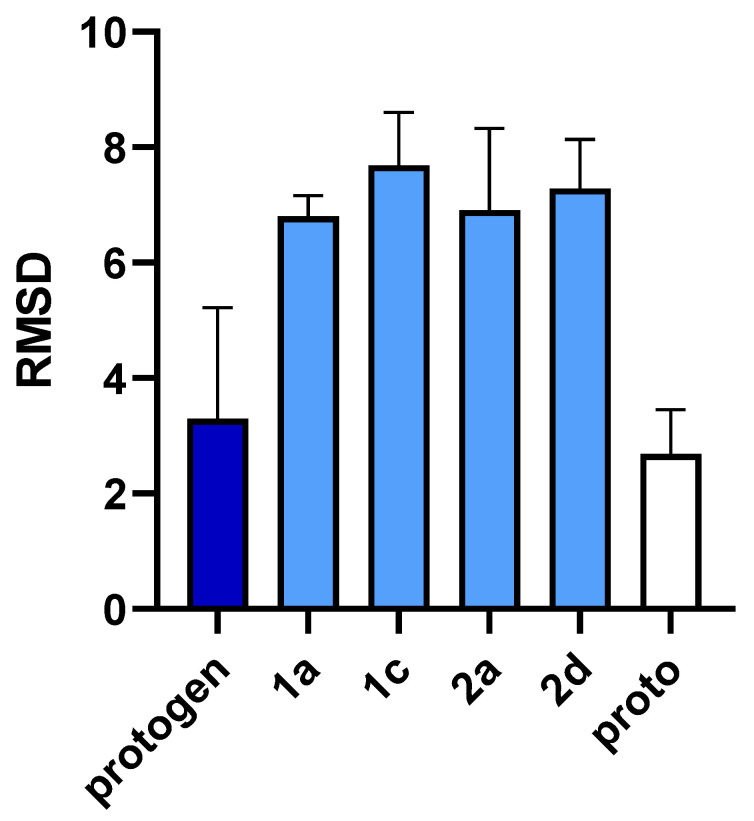
RMSD of the clusters with best poses for each tetrapyrroles relative to the conformations of the starting conformers prior to docking. Note that the starting conformers for protogen, 1a, 1c, 2a and 2d were designed using the coordinates of coproporphyrinogen bound within uroporphyrinogen carboxylase (1R3Y) [8], whereas the starting conformer of proto is based on the published planar coordinate of this molecule [14].

**Figure 5 ijms-21-09495-f005:**
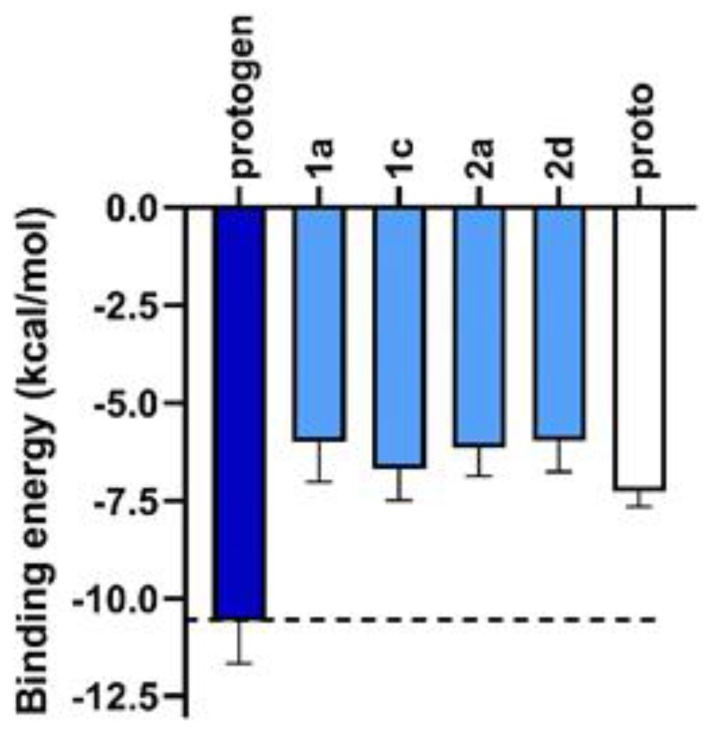
Binding energy (kcal/mol) of protogen (dark blue bars), reactions intermediates (light blue bars) and proto (white bars) to human PPO. The more negative the binding energy reflects a better fit within the catalytic domain. Dotted line represents the average binding energy of protogen.

**Figure 6 ijms-21-09495-f006:**
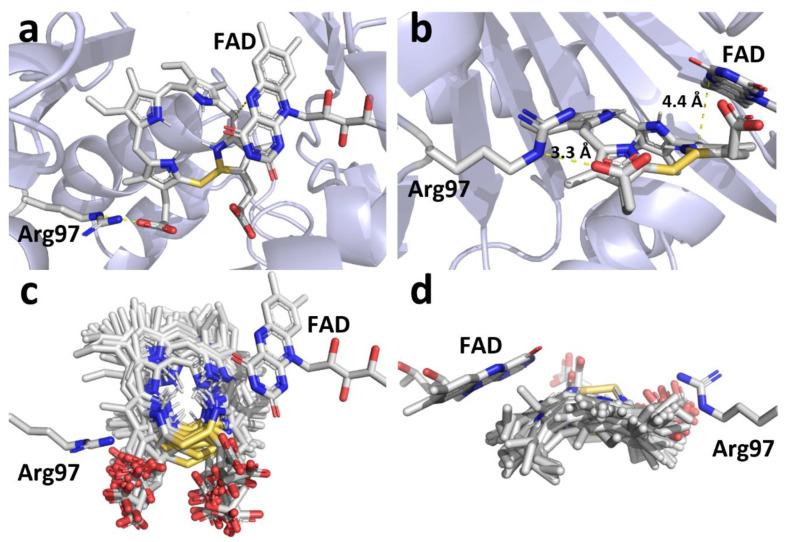
Docking of protogen to PPO using Autodock with Lennard-Jones forces. Similar results were obtained with other reaction intermediates (Appendix A). View of the docked protogen with the most favorable binding energy (**a**) from above and (**b**) from the opening of the catalytic domain. All of the top docking orientation group are shown to indicate flexibility within the pocket from above (**c**) and from the back of the pocket (**d**). Yellow dotted lines in (**a**) and (**b**) represent the interactions between protogen and PPO. The average distances between the N5 of FAD and C10 of protogen and between the propionate group of ring D and the Arg97 of PPO are shown in panel (**b**). Gold bonds represent the location of the Lennard-Jones forces between two dummy atoms.

**Figure 7 ijms-21-09495-f007:**
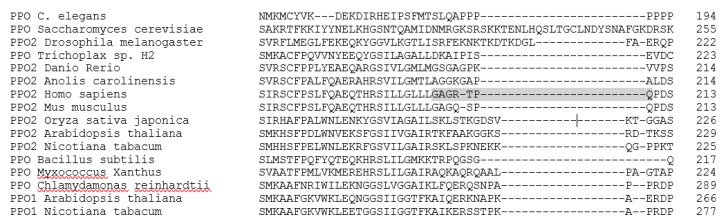
Alignment of 16 representative PPO sequences from model organisms with well characterized genomes, in the region of the unstructured loop. The loop from the human structure is highlighted.

**Table 1 ijms-21-09495-t001:** Summary of the parameters associated with the molecules docked in the expected orientation in PPO.

	Selected Docked Molecules ^1^	Distance between N5 of FAD and C10 of Tetrapyrrole (Å)	Distance between Propionate Group and Arg97 (Å)
Protogen	23	4.4 ± 0.5	3.3 ± 0.8
1a	22	4.3 ± 0.7	3.5 ± 0.9
1c	12	4.3 ± 0.5	3.3 ± 0.5
2a	17	4.5 ± 0.6	3.7 ± 1.3
2d	20	4.3 ± 0.5	4.1 ± 1.7
Proto	26	4.6 ± 0.2	4.1 ± 0.8

^1^ Total # of docked molecules: *n* = 50.

**Table 2 ijms-21-09495-t002:** Distance between Summary of the parameters associated with the molecules docked in the expected orientation in PPO.

	Distance between C10 Methylene H and N5 of FAD (Å) ^1^		Distance between Methylene H and Pyrrole N (Å) ^1^	
	α-H	β-H	α-H	β-H
Protogen	3.7 ± 0.8	5.1 ± 0.7	-	-
1a	-	-	3.0 ± 0.1	3.3 ± 0.1
1c	3.6 ± 0.6	5.0 ± 0.8	-	-
2a	-	-	2.9 ± 0.2	3.3 ± 0.1
2d	3.6 ± 0.6	5.0 ± 0.7	-	-

^1^ Refer to Figure 1 for position of α-H and β-H on C10 in protogen and intermediates 1c and 2d involved in the hydride abstraction, and on the methylene carbons on intermediates 1a and 2a involved in initiating the tautomerization restoring the sp3 state of C10 prior to the next oxidation.

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
