# Peer review of "Conformation of the Intermediates in the Reaction Catalyzed by Protoporphyrinogen Oxidase: An In Silico Analysis"

_ijms, 2020, doi:10.3390/ijms21249495_

Round 1

Reviewer 1 Report

The results of the paper illustrate the fact that a proposed mechanism may be correct which is noteworthy. There must be a way to accomplish this transformation but I am not convinced that this paper adds much to what is already known. The main conclusion appears to be that the alpha H atom is closer to FAD than the beta one which was already known. The paper may be suitable for publication after these questions below are addressed.  The paper also lacked the supporting information and some appendix A which may have resulted in an easier appreciation of its merits.

  1. The abstract should only state the results.
  2. Not sure about this numbering scheme and even the use of letters. The molecule depicted in Ref 7 is different from that displayed in Fig 1.  Which one is correct?
  3. Is there a hydride abstraction, i.e., H- or a proton, H+, or a hydrogen atom with FAD. If it is a hydride as stated then this would lead to HFAD‑ and the intermediates should be depicted with a negative charge.  On line 69 we have FADH going to FAD without the charges indicated.  Obviously, if it were the abstraction of a hydrogen atom, then FADH would be produced.
  4. In ref 7, the first H atom removed is on the methylene bridge between C and D. Here we have the first one being removed from A and B.  Are there not different energetic requirements for this removal and which one is correct?
  5. This protogen molecule could be oriented within PPO in two orientations with the ARG97. Is it possible for it to depart and then re-enter in a different orientation and do H atom abstraction from the other side?
  6. 7 refers to an Arg 98. What is responsible for this difference?  Line 76 does relate to ARG98.  Are these different forms of PPOs?
  7. Would the insertion of an Mg2+ or maybe an interstitial hydronium ion interacting with the lone pairs on the N atoms assist with this transformation? Can this be considered?
  8. Were there any conformational changes between the minimized structures and that for the copro?
  9. Not clear if there is an Appendix A or supplementary material (line 73) and/or this needs to be removed from the journal template. I could not find the supporting info, see line 213-214.
  10. Free protogen would have a very twisted structure as the H atoms attached to the N atoms in the middle would try to minimize steric hindrance. The starting structure in Fig. 2 which was docked into PPO may not be correct. Is protogen synthesize in situ within this PPO enzyme?  Only then would Fig. 2 be possible perhaps with external interactions to maintain that conformation which is more “planar” than that shown below.
  11. On line 112, flexible or weakest? If weakest estimate by how much if known.
  12. Was the model in Fig 3 with the dummy atoms (G) used in the calculations and is that why the model in Fig.1 removed the H atom between A and B initially in contrast to the mechanism in Ref 7? Fig. 5 would indicate that G was preserved in the calculations. Is this not a problem with the mechanism?
  13. Line 125. Could not locate Sup Fig 2.
  14. Very difficult to appreciate the nature of the structure. Would be helpful to have the data to twirl the model around. Fig.5 depicts an almost planar arrangement which would be impossible if the H atoms were still bonded to the central N atoms.  This is not believable.
  15. What were the conformation changes when protogen morphed through the various intermediates in Fig.1? Surely there would be major changes in the conformation.
  16. Please comment on this paper which would appear to address a similar topic.

Hao G-F, Tan Y, Yang S-G, Wang Z-F, Zhan C-G, Xi Z, et al. (2013) Computational and Experimental Insights into the Mechanism of Substrate Recognition and Feedback Inhibition of Protoporphyrinogen Oxidase. PLoS ONE 8(7): e69198. https://doi.org/10.1371/journal.pone.0069198

  1. On line 222 was the structure obtained from “3nks [26]?” Is this correct?  [26] is a docking program.

Author Response

  1. Via the journal website: “Abstract: The abstract should be a total of about 200 words maximum. The abstract should be a single paragraph and should follow the style of structured abstracts, but without headings: 1) Background: Place the question addressed in a broad context and highlight the purpose of the study; 2) Methods: Describe briefly the main methods or treatments applied. Include any relevant preregistration numbers, and species and strains of any animals used. 3) Results: Summarize the article's main findings; and 4) Conclusion: Indicate the main conclusions or interpretations. The abstract should be an objective representation of the article: it must not contain results which are not presented and substantiated in the main text and should not exaggerate the main conclusions.” Therefore inclusion of conclusions in the abstract follows the journal guidelines.
  2. The figure legend has been updated to specify that the figure is an adaptation from figure 6 of reference #7, the molecules are indeed the same. Thank you for the catch on the nomenclature error in the ring labeling of the tetrapyrrole, we have corrected this in Figure 1 and throughout the text where appropriate to agree with the past consensus.
  3. The reviewer is correct. An FADH- is created because the reaction is indeed a hydride extraction. The text has been corrected in three locations.
  4. In reference 7 Figure 6 the first hydride removed is between A and D. With our corrected labeling the first hydride is removed from the same location in the catalytic domain, but is between B and C. As far as whether there are different energetic requirements for the removal of the hydride, the removal is only possible from the hydrogen on the methylene bridge proximal to the N5 of FAD.
  5. The reviewer brings up an interesting question. As far as we know there is no published evidence of dissociation and reassociation occurring during the reaction. This may be due to the fact that the intermediates are likely to be unstable requiring the reaction to proceed rapidly.
  6. I have changed line 76 to agree with the rest of the paper. Arg98 of the tobacco protein (1sez) is equivalent to Arg97 of the human protein (3NKS).
  7. We are not sure what the reviewer is asking. Chelation with a 2+ metal occurs in this pathway and is catalyzed by separate chelatase enzymes which use the fully oxidized protoporphyrin IX so we don’t understand how a magnesium would contribute to the reaction catalyzed by PPO. The potential participation of a hydronium ion in the interaction has not been considered in our study.
  8. There were no significant changes in the minimized structures for copro or the intermediates. We have added an RMSD analysis to the paper to address this point.
  9. Our supplementary material was apparently not submitted correctly, we apologize and will rectify. The table in question is a percent identity matrix from a clustal omega alignment.
  10. You are correct, free proto does orient itself in a more “tent-like” structure due to the repelling of the hydrogen atoms in the center of the rings. I have updated the results and discussion to clearly explain how we addressed this in our methodology. Protogen is synthesized in coproporpyrinogen III oxidase which has been shown to be associated in situ with protoporphyrinogen IX oxidase and lies in the pathway between uroporphyrinogen oxidase (used to create the starting conformation) and PPO. All three proteins have a similarly sized catalytic domains. It was fortunate that the structure of coproporphyrinogen bound in uroporphyrinogen oxidase was captured in a crystal structure so we can be confident that the protons in the center of the tetrapyrrole can exist at this level of proximity. Furthermore, it turns out that protogen in this conformation also fits within the PPO active site without any difficulty. This is why our starting conformation is reasonable.
  11. It truly is the most flexible and not the weakest. We have updated the paper to reflect the reasoning behind the “most flexible bond”. Also the incorrect reference to figure 3 at the end of the sentence was corrected to figure 1
  12. I think this goes back to point #4, the hydrogen removed first is between B and C. Regarding the question about the dummy atoms, the answer is yes the dummy atoms were preserved during the calculations. This was necessary to allow the tetrapyrrole rings to be flexible during docking experiments. The dummy atoms do not change the final binding energies.
  13. See #9.
  14. The reviewer makes a good point, the figure is “busy” but is the best we can do with a static figure and we wanted to include it for the readers to appreciate that the conformation of the bound substrates in multiple iterations had very similar binding orientations. We have also added the 3d structure file (.pse) as a supplemental material. As far as the structure of protogen in the catalytic domain of the protein: the reason we believe the conformation of this substrate is correct is because the bond angles in these structures are very similar to those observed in coproporphyrinogen with all of the H atoms still bonded to the central N atoms when bound in uroporphyrinogen oxidase. It is also a well established theorem that enzymes hold substrates in high energy states to facilitate faster reactions than can happen without enzymes. Finally, some of the docking orientations found in our study were similar to those shown in the Hao et al paper (mentioned by the reviewer), with a more bent orientation, but these were not among those with the lowest binding energies.
  15. The reviewer makes a valid point, we do not show the intermediates bound to the catalytic domain in the paper for sake of simplicity. There were minor structural changes as the enzyme holds the substrate and intermediates in a general place, and the introduction of each additional double bond creates a more planar molecule until the fully conjugated and planar protoporphyrin IX. We will add figures of the intermediates in the supplemental materials.
  16. Thank you for the reference to the paper we missed. It has been added to the paper. It is on a similar subject, but rather tangential and our results do agree with the analysis within. It is good to see that newer methods do agree with their analysis.
  17. The reference I have on that appears to be correct: Qin, X.; Tan, Y.; Wang, L.; Wang, Z.; Wang, B.; Wen, X.; Yang, G.; Xi, Z.; Shen, Y., Structural insight into human variegate porphyria disease. FASEB J. 2011, 25, (2), 653-664.

Reviewer 2 Report

I recommend acceptance of the manuscript in its present form. I read carefully the manuscript and the message the authors want to convey is quite clear to me

Author Response

Reviewer did not require any changes

Reviewer 3 Report

Here are my comments regarding this manuscript before acceptance:

  1. the novelty of the paper needs to be better described at the end of the introduction.
  2. a distinctive and informative conclusion is needed.
  3. the language throughout the paper needs to be edited. some logical flow issues can be detected.

Author Response

  1. We have updated the last paragraph of the introduction
  2. A conclusion section has been added
  3. Minor changes to some sentences have been made throughout to unify voice

Round 2

Reviewer 1 Report

The comments are hopefully attached as an image
